# Disentangling Relationships among the Alpine Species of *Luzula* Sect. *Luzula* (Juncaceae) in the Eastern Alps

**DOI:** 10.3390/plants12040973

**Published:** 2023-02-20

**Authors:** Špela Pungaršek, Jasna Dolenc Koce, Martina Bačič, Michael H. J. Barfuss, Peter Schönswetter, Božo Frajman

**Affiliations:** 1Slovenian Museum of Natural History, Prešernova 20, SI-1000 Ljubljana, Slovenia; 2Department of Botany, University of Innsbruck, Sternwartestraße 15, A-6020 Innsbruck, Austria; 3Department of Biology, Biotechnical Faculty, University of Ljubljana, Jamnikarjeva 101, SI-1000 Ljubljana, Slovenia; 4Department of Botany and Biodiversity Research, University of Vienna, Rennweg 14, A-1030 Vienna, Austria

**Keywords:** agmatoploidy, AFLP fingerprinting, chromosome number, polyploidisation, relative genome size

## Abstract

Polyploidisation, agmatoploidy and symploidy have driven the diversification of *Luzula* sect. *Luzula*. Several morphologically very similar species with different karyotypes have evolved, but their evolutionary origins and relationships are unknown. In this study, we used a combination of relative genome size and karyotype estimations as well amplified fragment length polymorphism (AFLP) fingerprinting to investigate the relationships among predominately (sub)alpine *Luzula alpina*, *L. exspectata, L multiflora* and *L. sudetica* in the Eastern Alps, including also some samples of *L. campestris* and *L. taurica* as outgroup. Our study revealed common co-occurrence of two or three different ploidies (di-, tetra- and hexaploids) at the same localities, and thus also common co-occurrence of different species, of which *L. sudetica* was morphologically, ecologically and genetically most divergent. Whereas agmatoploid *L. exspectata* likely originated only once from the Balkan *L. taurica*, and hexaploid *L. multiflora* once from tetraploid *L. multiflora*, the AFLP data suggest multiple origins of tetraploid *L. multiflora*, from which partly agmatoploid individuals of *L. alpina* likely originated recurrently by partial fragmentation of the chromosomes. In contrast to common recurrent formation of polyploids in flowering plants, populations of agmatoploids resulting by fission of complete chromosome sets appear to have single origins, whereas partial agmatoploids are formed recurrently. Whether this is a general pattern in *Luzula* sect. *Luzula,* and whether segregation of ecological niches supports the frequent co-occurrence of closely related cytotypes in mixed populations, remains the subject of ongoing research.

## 1. Introduction

Polyploidy is one of the most important evolutionary pathways in flowering plants and has significantly contributed to their diversification and radiation [1,2,3,4]. Multiple and recurrent formation of polyploids, (epi)genetic, transcriptomic and genomic changes as well as morphological, geographic and ecological divergence following polyploidisation are considered significant processes in the evolution of polyploids [5,6,7,8,9,10]. Whereas autopolyploids arise within a single population or among ecotypes of a single species, allopolyploids are derivates of interspecific hybrids [11]. Especially autopolyploidy often generates multiple ploidy levels within single species and such intraspecific polyploids are commonly recurrently formed [12].

Contrary to true polyploidy, where the genome size is expected to increase proportionally to the increase in ploidy level (but can later again decrease due to genome downsizing; [13,14]), there are also cases where the increase in chromosome number is not accompanied by a change in genome size [15,16]. This phenomenon is named agmatoploidy [17,18]) and is based on fission of holocentric (holokinetic) chromosomes. In contrast to monocentric chromosomes, where spindle microtubules attach to a single kinetochore during mitosis or meiosis, in holocentric chromosomes, they attach along the whole length through diffuse kinetochores [19]. In addition, holocentric sister chromatids are interconnected along their whole length before anaphase disjunction and their fragmentation is, thus, not necessarily deleterious, as the fragmented parts are successfully segregated in daughter nuclei [19,20,21,22,23,24]. Contrary to agmatoploidy, a concerted fusion of holocentric chromosomes, named symploidy, can generate a karyotype with only half of the original chromosomes [25]. In addition to complete agmatoploidy and symploidy, also partial agmatoploidy and symploidy have been observed, with only a subset of chromosomes in the nucleus fragmented or fused, resulting in a continuous series of chromosome numbers [19,24,26].

Both true polyploidy as well as agmatoploidy and symploidy have driven the diversification of *Luzula* sect. *Luzula* (Juncaceae). The fission of chromosomes led to three karyotypes characterized by differently sized chromosomes, namely full-size chromosomes (AL-type), half-size chromosomes (BL-type) and quarter-size chromosomes (CL-type). Due to partial agmatoploidy or hybridization, combinations between these basic karyotypes are also possible [16,20,27]. In the Alps, *Luzula* sect. *Luzula* is represented by eight taxa, whose distribution in the Eastern Alps has recently been revised based on an extensive herbarium revision combined with genome size and karyotype estimation [28]. However, the differentiation among most of the high-elevation species was intricate due to their high morphological similarity and partly overlapping ecology. Four species can be found above the timberline in the subalpine and the alpine belt of the Eastern Alps, namely *L. alpina* (karyotype 12 AL + 24 BL), *L. exspectata* (24 BL), *L. multiflora* (24 AL, 36 AL) and *L. sudetica*, (48 CL) [28,29].

With the exception of *L. sudetica* that is divergent both morphologically, having small seeds with very short caruncles and relatively small flower parts with dark tepals and capsules, and ecologically, growing in moist meadows and bogs in the montane and alpine zones [28,30], the other three species are very similar in habit and difficult to distinguish morphologically. They differ in minor quantitative traits, which are very variable [27,28,29,30,31]. Whereas the genome size data can be used to distinguish diploid *L. exspectata* (along with *L. sudetica*) as well as the hexaploid populations of *L. multiflora* from tetraploid *L. alpina* and *L. multiflora*, only karyological investigations can reliably distinguish the last two taxa [28]. In addition, bedrock preference can also aid species identification. *Luzula alpina* and *L. sudetica* prefer siliceous substrate but can also be found on limestone on locally acidic or, in the case of *L. sudetica*, wet soils. On the other hand, *L. exspectata* predominantly grows on limestone, but can occasionally be found on siliceous bedrock, and *L. multiflora* thrives on both substrates. All four species appear to be widespread in the Eastern Alps, with *L. alpina* and *L. sudetica* being most common in the Central Alps and *L. exspectata* in the Southern Alps [28]. A few reports of *L. exspectata* in the Northern Alps [28] were based solely on herbarium revision and it remains unknown whether the populations in the Northern Alps have a common origin with those from the Southern Alps, from where the species was described [16].

Little is known about the origin of and the phylogenetic relationships among the four species, as no phylogenetic studies providing sufficient resolution among the species of *L.* sect. *Luzula* are available to date. In a phylogenetic study of *Luzula* based on plastid *trnL–F* and internal transcribed spacer (ITS) sequences [23], relationships among the Alpine species and most other members of *L.* sect. *Luzula* remained unresolved. It is well known that in flowering plants, polyploids can be formed recurrently within single species [12], but nothing is known about the formation of agmatoploids. It is unclear whether chromosome fissions occur multiple times leading to recurrent formation of BL or CL karyotypes from different populations of the same parental species, or if such events are rare, rendering species such as *L. exspectata* and *L. sudetica* monophyletic. Bačič et al. [28] hypothesized that *L. exspectata* (24 BL) might have originated via chromosome fission from *L. taurica* (V. I. Krecz) Novikov, a species with 12 AL chromosomes, distributed between the Balkan Peninsula and the Caucasus [30,31]. In the same way, *L. sudetica* (48 CL) could have originated from *L. exspectata* [28]. Kirschner [26] suggested that the alpine tetraploid *L. multiflora* (24 AL) could be of autopolyploid origin, whereas the lowland hexaploid populations (36 AL) are probably of allopolyploid origin. Likewise, *L. alpina* (12 AL + 24 BL) could be of allopolyploid origin [26], with *L. exspectata* (24 BL) and *L. taurica* (12 AL) acting as putative parents [16,28,29]. Alternatively, based on strong morphological and ecological similarity with tetraploid *L. multiflora*, we hypothesized that *L. alpina* could be a partial agmatoploid, which originated by the fission of half of the chromosomes of *L. multiflora*.

Taking into account the unclear evolutionary history, the aim of the present study was to shed light on the relationships among the four alpine species of *Luzula* sect. *Luzula* based on a comprehensive geographic sampling of almost 440 individuals across the Eastern Alps. We applied ploidy level estimation via flow cytometric measurements of relative genome size (RGS) to discriminate among diploid *L. exspectata* and *L. sudetica*, tetraploid *L. alpina* and *L. multiflora*, and hexaploid *L. multiflora*. In addition, we estimated the chromosome number of several individuals in order to discriminate between tetraploid *L. alpina* and *L. multiflora* and to establish the karyotype of the diploids from the Northern Alps. Finally, we used amplified fragment length polymorphisms (AFLP) to explore the relationships among the four species. Specifically, we investigated whether polyploid and agmatoploid populations have single or multiple origins.

## 2. Results

### 2.1. Karyotype Variation

Cytogenetic analyses of 15 individuals revealed two diploid karyotypes, i.e., 12 AL of *L. taurica* (one individual) and 24 BL of *L. exspectata* (three individuals), as well as two tetraploid karyotypes, i.e., 24 AL of *L. multiflora* (seven individuals) and 12 AL + 24 BL of *L. alpina* (four individuals; Figure 1, Appendix A).

### 2.2. Relative Genome Size and Ploidy Levels

RGS was discretely distributed (Figure 2). Diploids had RGS between 0.22 and 0.27, tetraploids between 0.44 and 0.57, and hexaploids between 0.75 and 0.82. One individual of *L. multiflora* had RGS 0.65 and was, thus, putative pentaploid. Among the diploids, *L. taurica* had slightly lower RGS between 0.22 and 0.25, whereas there was no clear divergence of RGS between *L. exspectata* and *L. sudetica* that had their RGS scattered between 0.25 and 0.27; both individuals of *L. campestris* had their RGS in the range of these two species. Among the tetraploids, nine individuals from the westernmost localities 1 and 2 had clearly lower RGS (0.44–0.46), whereas individuals from all other localities, including four individuals from locality 1 and five from locality 2 had higher RGS ranging from 0.52 to 0.54. RGS values of the karyotyped individuals of *L. alpina* and *L. multiflora* were scattered across the entire higher range of tetraploid RGS variation, indicating no divergence in RGS between the two species. Among the hexaploids, two individuals had slightly lower RGS (0.74 and 0.77), whereas all others had their RGS ranging from 0.78 to 0.82. One individual from locality 2 had an RGS of 0.66, intermediate between the tetraploids and hexaploids, suggesting that it may be pentaploid. Besides this odd-ploidy cytotype, at the same locality, eleven individuals were tetraploid, four were hexaploid and two were diploid.

At several localities, two or even three ploidy levels were detected (Figure 3). Often, diploid individuals of *L. exspectata* and/or *L. sudetica* were intermingled with tetraploid individuals of *L. alpina*/*L. multiflora* at a small spatial scale; occasionally, single hexaploid individuals of *L. multiflora* were growing at the same localities (Figure 4).

### 2.3. AFLP Fingerprinting and Analyses of AFLP Data

Three hundred and eighty-one AFLP fragments were scored in 306 individuals from which high-quality, reproducible AFLP fingerprints were obtained; 29 fragments were found in only one individual and were excluded from further analyses. The average replicate error rate was 3.51% for all individuals and 3.75% for tetraploids. The NeighbourNet (Figure 5a) of all samples was star-like, but several clear clusters sharing multiple parallel splits were revealed. All diploid individuals clustered together: the clearest cluster with longest split included *L. sudetica*, one less clear cluster included *L. exspectata* and slightly divergent *L. taurica*, from which two individuals of *L. campestris* were also clearly divergent. On the other hand, the tetraploid populations of *L. alpina/L. multiflora* were highly variable and formed several divergent clusters, with karyotyped individuals (i.e., *L. alpina* and *L. multiflora*) scattered across them. Finally, all hexaploid individuals formed a separate cluster positioned within the tetraploids. The PCoA revealed similar results, with diploids and hexaploids being most divergent (Figure 5b). The first two axes explained 17.8 and 7.5% of the total variation, respectively.

In accordance with the results of the entire dataset, the NeighbourNet and the PCoA of the tetraploid individuals indicated two main groups of individuals, of which one was more compact in the PCoA and the other clearly more scattered (Figure 6a,b). The first two axes explained 16.2 and 7.5% of the total variation, respectively. Non-hierarchical K-means clustering revealed two or four optimal clusters according to Evanno et al. [32]. The two clusters corresponded well with the two main clouds in the PCoA. At K = 4, four individuals were split from the more compact cloud, and, likewise, the most divergent individuals were separated from the more dispersed cloud. The genetic variation was organized in west–east direction (Figure 6c), the westernmost populations being most divergent (Figure 6d). Interestingly, within the admixed populations, the individuals belonging to the two genetic clusters were geographically segregated (Figure 6e–h). One of the two main genetic clusters included two karyotyped individuals of *L. alpina* and one of *L. multiflora,* and the other two of *L. alpina* and six of *L. multiflora*.

## 3. Discussion

### 3.1. Monophyletic Origin of Agmatoploid L. exspectata and L. sudetica

Contrary to the commonly recurrent formation of polyploids [28], our results clearly indicate that agmatoploid *L. exspectata* (2*n* = 24 BL) and *L. sudetica* (2*n* = 48 CL) both have a single origin. This may suggest that fission of chromosomes leading to different agmatoploid karyotypes is not a common process. In the absence of comparable large-scale studies including multiple populations in both *Luzula* as well as other genera with incidence of agmatoploidy [19,24,28,33]; generalizations are clearly premature.

Our AFLP data (Figure 5) showed that *L. sudetica* is not just morphologically and ecologically [28], but also genetically, most divergent from the other investigated species. Bačič at al. [28] hypothesized that the 48 CL karyotype of *L. sudetica* originated once via chromosome fission from *L. exspectata* (2*n* = 24 BL), and shared splits between these two taxa are in support of this hypothesis. However, the sampling of additional species with the 24 BL karyotype such as *L. divulgatiformis* Bačič & Jogan would be needed to further test this hypothesis. Likewise, the origin of *L. exspectata* from the Balkan *L. taurica* (2*n* = 12 AL) via chromosome fission suggested by Bačič et al. [28] is supported by our data, as both species share multiple common splits in the NeighbourNet (Figure 5a) and are positioned close to each other in the PCoA scatterplot (Figure 5b). On the other hand, widespread and largely sympatric *L. campestris* (2*n* = 12 AL) appeared more distant. Finally, *L. pallescens*, the third diploid with 12 AL karyotype that could have acted as a parent of *L. exspectata*, was not included in our study; further studies are therefore needed to finally resolve the origin of *L. exspectata*. Both karyotype (Figure 1) and genetic (Figure 5) data, however, clearly showed that the putative populations of *L. exspectata* from the Northern and Central Alps previously identified solely by morphology [28] are clearly conspecific with those from the Southern Alps, from where *L. exspectata* was described [16].

### 3.2. Recurrent Formation of Tetraploids and Agmatoploids in the L. alpina/L. multiflora Complex

Contrary to the diploid species and the hexaploid populations of *L. multiflora* that were all genetically relatively homogeneous, the tetraploids were highly variable and formed several divergent AFLP clusters (Figure 5), suggesting their multiple origins. This is not surprising, as the recurrent formation of polyploids is not rare (Soltis et al. 2016) and multiple origins of tetraploids from diploid progenitors have been evidenced in different Alpine plant groups, e.g., in *Biscutella laevigata* L. [34,35], *Lotus alpinus* (Ser.) Schleich. [36] and—more ambiguously—in the *Senecio carniolicus* Willd. group [37].

The position of *L. campestris* in the NeighbourNet (Figure 5) close to tetraploid *L. alpina*/*L. multiflora* suggests that *L. campestris* has likely been involved in the origin of tetraploid *L. multiflora*. As *L. campestris* is widespread across low elevations in the Alps [28], it appears plausible that different genetic clusters of *L. multiflora* originated independently via multiple whole-genome duplications from geographically close populations of *L. campestris*. Kirschner [26] suggested that the alpine tetraploid *L. multiflora* subsp. *multiflora* is of autopolyploid origin derived from *L. pallescens* (12 AL), that we did not include in our study, but since the latter species does not occur in the Eastern Alps, this hypothesis appears less plausible.

For *L. alpina*, Kirschner [26] suggested an allopolyploid origin and Bačič et al. [16,28,29] hypothesized that *L. exspectata* (24 BL) and *L. taurica* (12 AL) could be its putative parents. However, genetic data in the present study do not support this hypothesis, as karyotyped individuals of *L. alpina* were scattered among karyotyped individuals of *L. multiflora* within different AFLP clusters (Figure 6). This indicates that *L. alpina* likely had multiple origins, and we here suggest that *L. alpina* originated several times independently from tetraploid *L. multiflora* via fragmentation of only half of the chromosomes and is thus of partial agmatoploid rather than allopolyploid origin. The marked morphological similarity between *L. alpina* and *L. multiflora* [16,27,28,30] also supports this hypothesis. Based on our measurements of diagnostic morphological characters (Appendix A; Appendix A), these two species appear to be indistinguishable (see also Appendix A). In addition, albeit only based on a limited number of karyotyped individuals, the two species grew in the same habitats, where they can be found in close vicinity (Figure 6e–h).

Partial agmatoploidy was also observed in other *Luzula* species; for example, in *L. spicata*, the most common chromosome number was 2*n* = 24 BL, but there were a few samples with 2*n* = 10 AL + 4 BL [20]. It has been suggested that prevalence of agmatoploidy is linked to various stressful habitats and that increased UV radiation at higher elevations can trigger fragmentation of holocentric chromosomes [38]. As tetraploid *L. multiflora* usually grows at higher altitudes above the timberline, the strong UV radiation could trigger occasional fission of chromosomes. However, it remains unclear if partly agmatoploid individuals of *L. alpina* can establish viable populations.

The non-hierarchical K-means clustering including all tetraploid individuals revealed geography-correlated variation of genetic differentiation in the west–east direction (Figure 6c). Similar longitudinally organized genetic patterns were uncovered in several silicicolous Alpine species and were explained by glacial survival in different Last Glacial Maximum refugia close to the southern and eastern margin of the Eastern Alps [39]. The westernmost populations were most divergent (Figure 6a,b,d), albeit morphologically similar to tetraploids from other localities (Appendix A). Interestingly, several, but not all, individuals from the westernmost populations 1 and 2 had divergent RGS, much lower than that of all other tetraploids (Figure 2). Further studies are needed to reveal whether RGS-divergent plants are more common in the western part of the Eastern Alps and the adjacent Western Alps. Finally, non-hierarchical K-means clustering indicated a striking segregation of genetic variability also at the local geographic scale (Figure 6e–g). It appeared unlikely that this segregation is related to habitat differentiation, but additional studies are needed to explore this further.

Unlike the tetraploids, the hexaploid populations of *L. multiflora* appear to have had a single origin from the tetraploid populations. Although Kirschner [26] suggested that the lowland hexaploid populations are probably of allopolyploid origin, our data rather suggest that they are autopolyploids, as they were all clearly nested within the tetraploids in the NeighbourNet (Figure 5a). Even if the hexaploids appear to have their elevational distribution centered at lower elevations [28], in the investigated localities, we found several hexaploid individuals scattered among tetraploid *L. alpina/L. multiflora* and occasionally also among diploid *L. exspectata* (Figure 4). It appeared that individuals of different ploidy levels, except *L. sudetica*, were fairly randomly distributed at a local scale (Figure 4), which suggests that they occupy similar ecological niches. Nevertheless, even if niche differentiation is considered the prevalent cause of cytotype separation, it is difficult to demonstrate without experimental comparisons of plant fitness in contrasting environments [40]. Such separation can be caused also by colonization history, limited dispersal and clonal expansion. Assortative mating in mixed-ploidy species can be achieved through multiple mechanisms including spatial segregation of cytotypes, asynchronous flowering and assortative pollen transfer [41]. The odd-ploidy pentaploid cytotype detected in one population (Figure 2) among dominant ploidies could either play an important role in polyploid evolution as a mediator of gene flow and recurrent polyploidisation, or, alternatively, be sterile, and thus, represent an evolutionary dead end [41].

## 4. Materials and Methods

### 4.1. Plant Material and Taxon Identification

We sampled 420 individuals from 27 locations of *L. alpina, L. exspectata, L. multiflora* and *L. sudetica* in the Eastern Alps (Austria, Italy and Slovenia). We also included two individuals of *L. campestris* (L.) DC. from two localities and eight individuals of *L. taurica* from a single locality (both 2*n* = 12 AL) as outgroups (Figure 3; Appendix A). In most cases, GPS coordinates of each individual were recorded; only in few cases when multiple individuals were collected on a very small patch, a single coordinate was recorded. Leaf material of three to ten individuals per locality and taxon was collected and immediately stored in silica gel for flow cytometric and AFLP analyses. In addition, at least one herbarium voucher was collected per locality and taxon, but in most cases, each silica-sampled individuum was herbarized. Vouchers are deposited at the herbarium IB.

The identification of all individuals was carried out as a combination of field observations regarding morphology and ecology, and ploidy-level estimations in the lab. In this way, diploid *L. exspectata* and *L. sudetica* as well as hexaploid *L. multiflora* could be reliably identified. On the other hand, the discrimination between tetraploid *L. alpina* and *L. multiflora* was mostly not possible, although we measured the most important characters that are supposed to discriminate between the two species, i.e., the width of the basal leaves and the length of the longest peduncle in the inflorescence [28,42] (Appendix A). We could thus reliably identify only a few tetraploid individuals, for which we established the karyotype, whereas we treated all remaining tetraploids as “*L. alpina*/*L. multiflora*”, especially since both karyotypes were established for different individuals from the same localities, indicating common co-occurrence of both taxa (Appendix A).

### 4.2. Karyotype Determination

For the karyological analyses, we collected seeds from 13 populations of *L. alpina, L. exspectata* and *L. multiflora* in the Eastern Alps and one of *L. taurica* from the central Balkan Peninsula during field excursions between 2016 and 2021 (Appendix A, Figure 3). Seeds were germinated and root tips fixed, Feulgen-stained and squashed to prepare microscopic slides as described by Bačič et al. [28]. The chromosome number was determined using an Axioscope MOT light microscope (Carl Zeiss, Jena, Germany) with a 63× oil immersion objective, CCD camera (Sony DXC-950P; Sony, Köln, Germany), frame grabber Matrox Meteor (Matrox, Dorval, Quebec, Canada) and computer with the image analysis software KS 400 v. 3.0 (Carl Zeiss, Jena, Germany). Photographs of the chromosomes were produced using AxioVision v. 4.8.2 (Carl Zeiss, Jena, Germany).

### 4.3. Relative Genome Size Estimation

Flow cytometry (FCM) of 4′,6-diamidino-2-phenylindole (DAPI)-stained nuclei was used to estimate the RGS following the procedure described by Suda and Trávníček [43]. The RGS was estimated using *Bellis perennis* (2C = 3.38 pg DNA; [44]) as a reference standard. Three to ten individuals per population (Appendix A) were analyzed. A CyFlow space flow cytometer (Symex Partec, Görlitz, Germany) was used to record the relative fluorescence of 3000 nuclei and FloMax software v. 2.11 (Sysmex Partec) was used to evaluate histograms and to calculate coefficients of variation (CVs) of the standard and sample peaks. The RGS was calculated as the ratio between the values of the mean relative fluorescence of the sample and the standard. The package ‘ggplot2′ for R 4.2.2 [45] was used to produce a scatter plot and boxplots of RGS.

### 4.4. AFLP Fingerprinting and Analyses of AFLP Data

Total genomic DNA was extracted from ca. 10 mg of dried tissue applying a CTAB protocol [46] with slight modifications [47]. Two to ten individuals per population from 28 locations/71 populations (at several locations several species were sampled), totaling 306 individuals, were used for AFLP analyses: 61 individuals from 20 populations of *L. exspectata*, 29/11 of *L. sudetica*, 174/27 of tetraploid *L. alpina/L. multiflora* (of which we determined the karyotype for four individuals of *L. alpina* and seven of *L. multiflora*), 32/10 of the hexaploid *L. multiflora*, as well as two individuals from two populations of *L. campestris* and eight individuals from one population of *L. taurica* as outgroup (Appendix A).

The AFLP procedure followed Vos et al. [48] with the modifications described by Frajman et al. [49]. An initial screening of selective primers using 12 primer combinations with three nucleotides was performed. The three final primer combinations for the selective PCR (fluorescent dye in brackets) were EcoRI (FAM)-ACA/MseI-CAC, EcoRI (VIC)-AGG/MseI-CTG, EcoRI (NED)-ACC/MseI-CAG. The selective PCR products (3.3 µL) were purified as described in Schönswetter et al. [47]. Then, 1 µL of the elution product was mixed with 10 µL Hi-Di formamide and 0.13 µL GeneScan 500 ROX dye size standard (Applied Biosystems, Foster City, CA, USA) and run on an ABI 3130xl Genetic Analyzer automated capillary sequencer (Applied Biosystems). Two blanks (DNA replaced by water) were included to test for contamination, and 30 samples were used as replicates between the two PCR batches to test the reproducibility of the technique.

Electropherograms were analyzed separately for the whole dataset and for the tetraploids with Peak Scanner v1.0 (Applied Biosystems) in the size range from 100 to 500 bp using the default peak detection method. Automated binning and scoring of the AFLP fragments were performed using RawGeno 2.0-2 [50] for R 3.3.2 [51] with the following settings: scoring range: 75–500 bp, minimum intensity = 100 relative fluorescens units (RFUs), minimum bin width = 1 bp and maximum bin width = 1.5 bp. Fragments with a reproducibility lower than 80% based on sample-replicate comparisons were eliminated. The error rate [52] was calculated as the ratio of mismatches (scoring 1 versus 0) over phenotypic comparisons in AFLP profiles of replicated individuals. Fragments with singular occurrences were eliminated.

SplitsTree4 v.12.3 [53] was used to produce a NeighbourNet based on uncorrected P-distances. A principal coordinate analysis (PCoA) based on a matrix of Jaccard distances generated with the R-package vegan [54] was performed using the program ape 5.0 [55]. Both NeighbourNets and PCoA scatter plots were produced for the complete dataset as well as for the tetraploid individuals. Additionally, for the 175 tetraploid individuals, non-hierarchical K-means clustering [56] was performed using a script of Arrigo et al. [57] for R 3.3.2 [51]. A total of 50,000 independent runs were performed (i.e., starting from random points) for each assumed value for K clusters ranging from 2 to 10. To select the best number of groups, the strategy proposed by Evanno et al. [32] was used. In addition, we also present the results for two groups, corresponding to the number of taxa within the sample. The geographic distribution of the genetic groups was visualized in ArcGIS.

## Figures and Tables

**Figure 1 plants-12-00973-f001:**
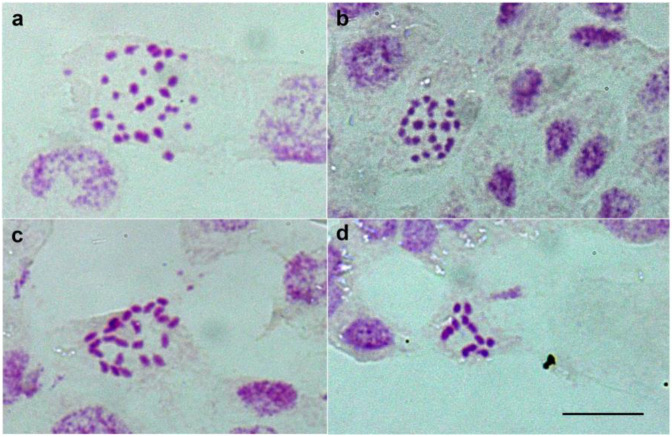
Metaphase or pro-metaphase chromosomes of (**a**) *Luzula alpina* (2*n* = 4*x* = 12 AL + 24 BL; location 12, individuum 106), (**b**) *L. exspectata* (2*n* = 2*x* = 24 BL; loc. 1, ind. 15808-105/7), (**c**) *L. multiflora* (2*n* = 4*x* = 24 AL; loc. 17, ind. 323) and (**d**) *L. taurica* (2*n* = 2*x* = 12 AL; loc. 28, ind. 15779-1). The bar represents 10 μm. Location numbers and individuum numbers correspond to Appendix A.

**Figure 2 plants-12-00973-f002:**
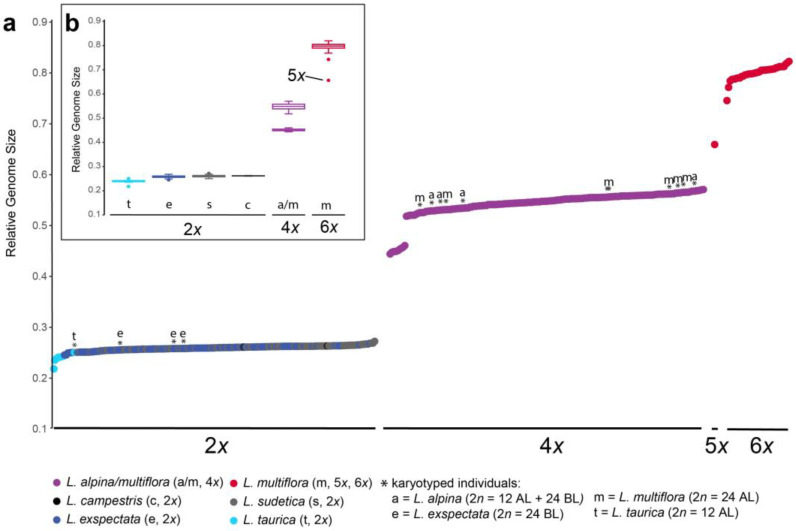
Relative genome size (RGS) of diploid (2*x*) *Luzula campestris*, *L. exspectata*, *L. sudetica* and *L. taurica*, tetraploid (4*x*) *L. alpina* and *L. multiflora* (24 AL), and hexaploid (6*x*) *L. multiflora*, including one pentaploid (5*x*) individual. (**a**) Scatterplot of all investigated individuals, ordered by increasing RGS; asterisks indicate karyotyped individuals labelled with the first letter of the epithet. (**b**) Box plots of RGS per species.

**Figure 3 plants-12-00973-f003:**
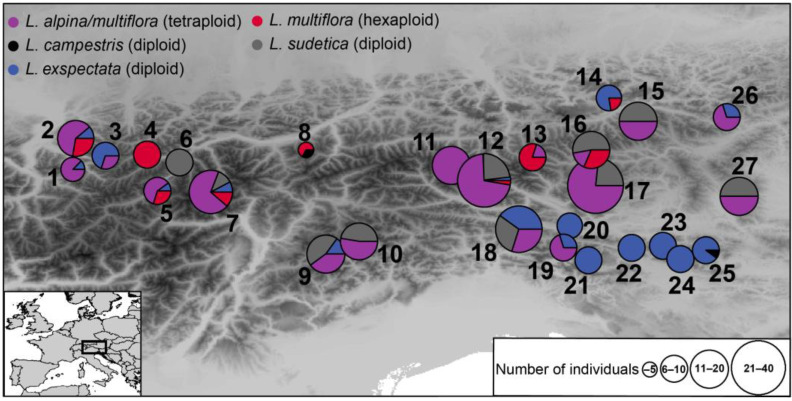
Sampling localities of *Luzula alpina/L. multiflora*, *L. campestris*, *L. exspectata*, *L. multiflora* and *L. sudetica*. Sizes of pie charts are proportional to the number of sampled individuals per locality; colors indicate the proportions of the sampled taxa. Locality numbers correspond to Appendix A.

**Figure 4 plants-12-00973-f004:**
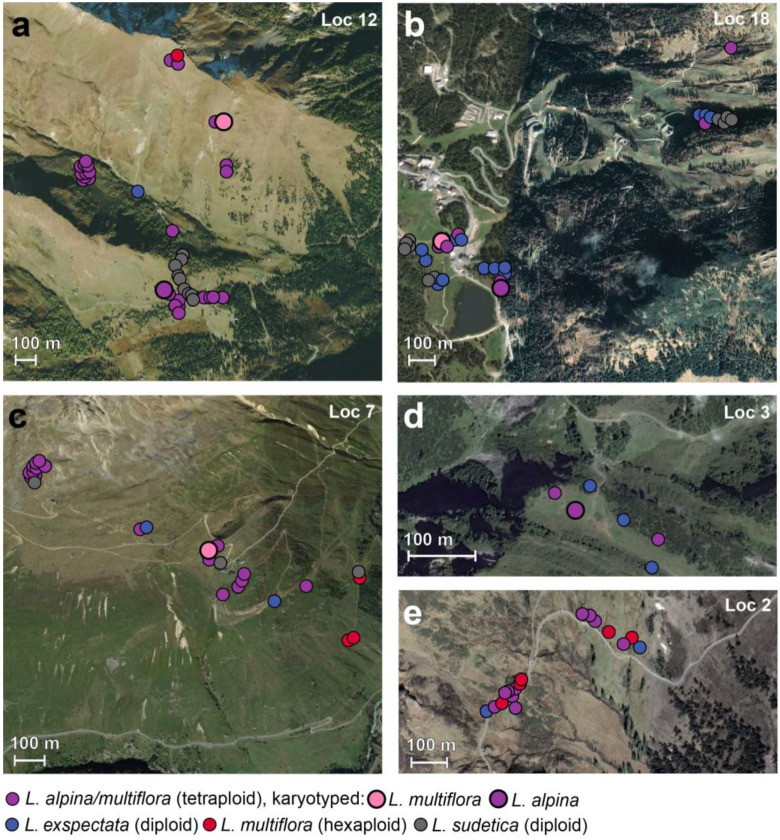
Small-scale geographical distribution of di-, tetra- and hexaploid individuals in selected ploidy-mixed localities. (**a**) Großfragant (Hohe Tauern), (**b**) Nassfeld (Karnische Alpen), (**c**) Vent (Ötztaler Alpen), (**d**) Batzigalpe (Lechquellengebirge), (**e**) Furkajoch (Bregenzerwaldgebirge). Colored dots represent the species with the ploidy level according to Figure 3. Bigger dots represent karyotyped tetraploid individuals. Location (Loc) numbers correspond to Appendix A.

**Figure 5 plants-12-00973-f005:**
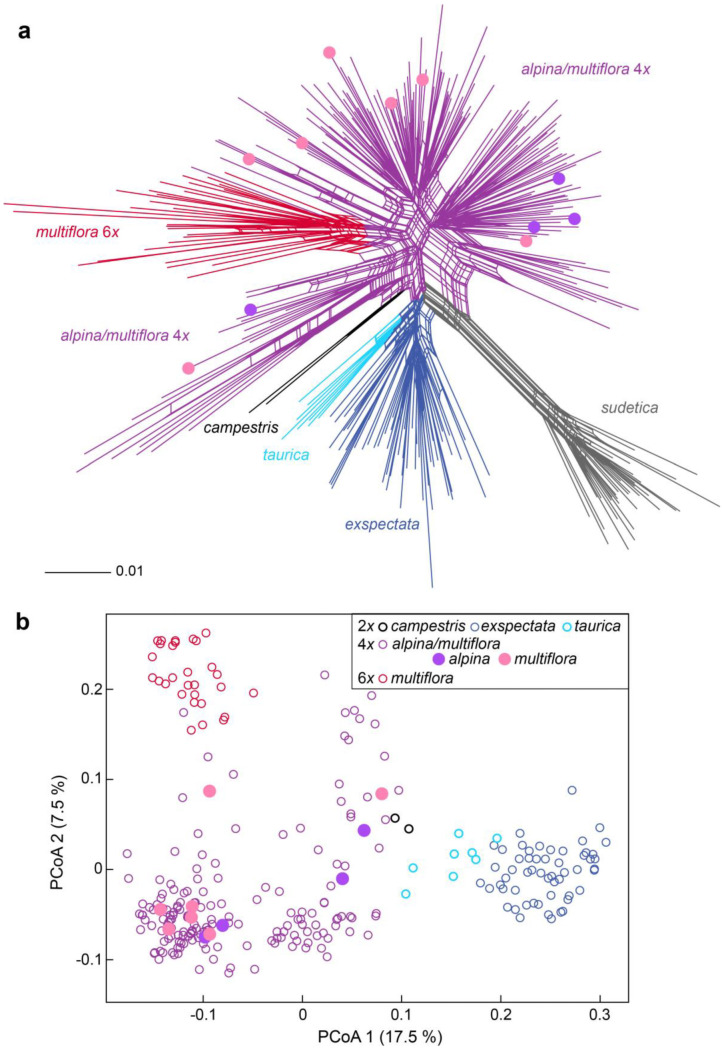
Genetic variation within mostly high-elevation Eastern Alpine species of *Luzula* sect. *Luzula* based on AFLP fingerprinting. (**a**) NeighbourNet; (**b**) Principal Coordinate Analysis scatterplot. Colors correspond to species/ploidy levels. Filled symbols indicate karyotyped individuals of *L. alpina* and *L. multiflora*.

**Figure 6 plants-12-00973-f006:**
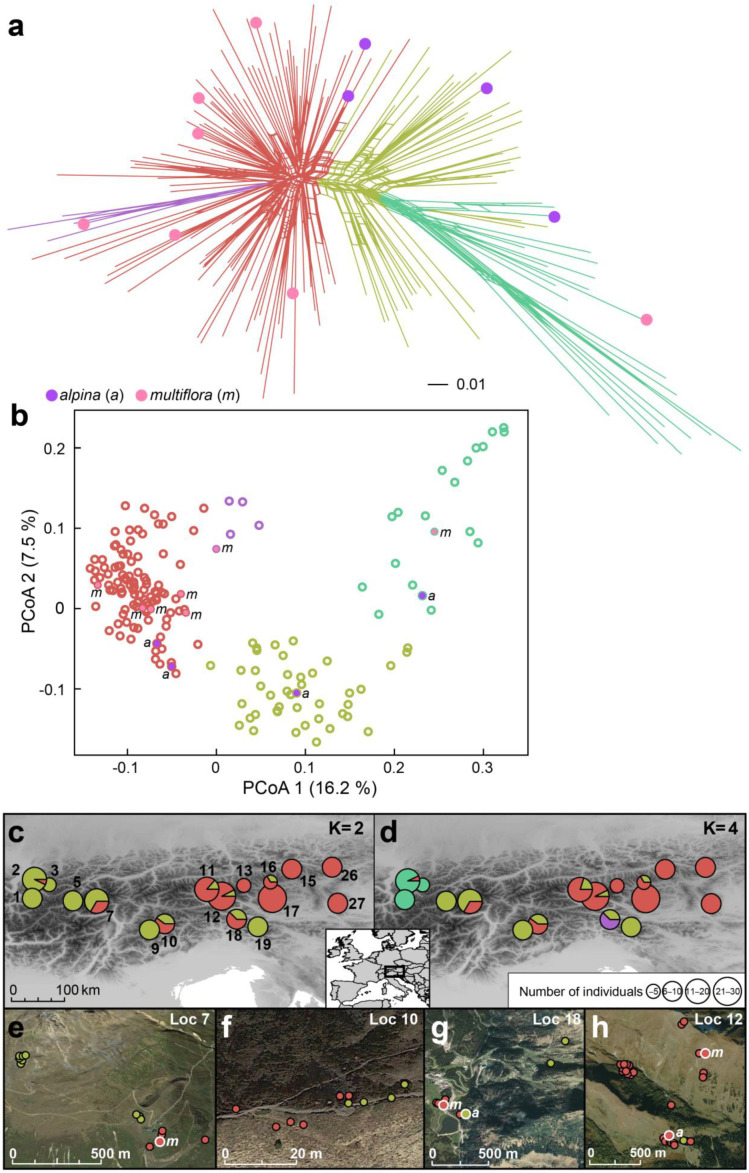
Genetic variation within high-elevation Eastern Alpine tetraploid populations of *Luzula alpina* and *L. multiflora* based on AFLP fingerprinting: (**a**) NeighbourNet, (**b**) Principal Coordinate Analysis scatterplot, (**c**) non-hierarchical K-means clustering for K = 2, and (**d**) K = 4. Colors in panels A and B indicate genetic clusters at K = 4. Karyologically identified individuals of *L. alpina* (violet filling, *a*) and *L. multiflora* (pink filling, *m*) are indicated. In panel (**c**), locality numbers correspond to Figure 3 and Appendix A. In panels (**c**) and (**d**), the sizes of the pie charts correspond to the number of investigated individuals as indicated in the inset in panel (**d**). In panels (**e**) to (**h**), the distribution of individuals from four populations with presence of both clusters at K = 2 is shown at the local scale and colors correspond to the genetic clusters in (**c**); dots with white margin indicate karyotyped individuals (*a, L. alpina*; *m*, *L. multiflora*). The locations (Loc) correspond to Figure 3 and Appendix A: (**e**) Vent (Ötztaler Alpen), (**f**) Passo Giau (Dolomiti), (**g**) Nassfeld (Karnische Alpen) and (**h**) Großfragant (Hohe Tauern).

## Data Availability

Data are contained within the article or the Appendix A.

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
