# Peer review of "Disentangling Relationships among the Alpine Species of *Luzula* Sect. *Luzula* (Juncaceae) in the Eastern Alps"

_plants, 2023, doi:10.3390/plants12040973_

Round 1
Reviewer 1 Report
Dear authors,
The aim of the manuscript “Disentangling relationships among the alpine species of Luzula sect. Luzula (Juncaceae) in the Eastern Alps” was to shed light on the relationships among the four alpine species of Luzula sect. Luzula based on a comprehensive geographic sampling of almost 440 specimens across the Eastern Alps.
The topic is original and address a specific gap in the field. Authors used a combination of relative genome size and karyotype estimations as well amplified fragment length polymorphism (AFLP) fingerprinting to investigate the relationships among Luzula alpina, L. exspectata, L multiflora and L. sudetica.
General concept comments: Authors sampled 420 individuals from 27 locations of L. alpina, L. exspectata, L. multiflora and L. sudetica in Austria, Italy and Slovenia. They also included two individuals of L. campestris (L.) DC. from two localities and eight individuals of L. taurica from a single locality. In most cases, GPS coordinates of each individual were recorded. Leaf material of three to ten individuals per locality and taxon was collected and stored in silica gel for flow cytometric and AFLP analyses. In addition, at least one herbarium voucher was collected per locality and taxon, but in most cases each silica-sampled individuum was herbarized. Authors wrote that vouchers were deposited at the herbarium IB.
Taking into account that the diagnosis of these four Luzula species is difficult, it is necessary to digitize the voucher herbarium specimens which were used for AFLP analysis with IB barcodes and publish the dataset in Global Biodiversity Information Facility (GBIF.org) or to place the digital images of Luzula voucher specimens into Virtual herbarium with links in the Table 1. In this case the results should be comparable, may be controlled, and used for the future research.
Author Response
We are currently continuing our studies on Alpine Luzula sect Luzula within a bigger project LuzAlp (https://www.uibk.ac.at/de/botany/forschung/projekte/same-same-but-different/), where we will also partly include the specimens from the present study. As we are also performing some morphometric analyses on the specimens, they were not yet mounted on herbarium sheets and are therefore not ready to be scanned yet. Herbarium IB is a public herbarium that is sending its collections on loan, therefore availability of specimens after the end of the LuzAlp project for further research activities of the scientific community is granted. In addition, the herbarium IB is part of an Austrian initiative within which the herbarium collections will be scanned in the future years. Nevertheless, since the species of L. sect. Luzula can only (partly) be distinguished in minor morphological characters connected to flowers and fruits (see Bačič et al., 2019), preparation of flowers is needed for their detailed study, therefore scanned images are of little use for further scientific research – see also the immense morphological variability and similarity among species presented in Supplementary Fig. 2. This said, we are currently not able to provide scanned images of the specimens.
Reviewer 2 Report
The study system is very interesting, however, I am concerned about the markers used. AFLPs are highly dependent of context, as any fission or fusion of chromosomes could change the genomic context of specific restriction sites, making the homology of bands impossible to determine. Simply put, these markers are not suitable for this particular group of species, especially without any alternative sequence information that could support the results from AFLP analyses. I suggest the authors try using chloroplast intervenir spacers or nuclear gene sequences to see if the patterns same relationships are recovered. APLPs are not very informative when trying to determine phylogenetic relationships in the best of cases, but in this context they are very unreliable.
Author Response
Given the clear structure inferred by AFLP analysis, where different populations and individuals of the same species group together, we do not believe that the fission of chromosomes could have played an important role in our AFLP analyses. Agmatoploid fission is something happening in the living cells and not in the DNA extract that is used for enzymatic restriction, therefore we see no problem in using AFLP fingerprinting in agmatoploid groups. Previously, phylogenetic studies using ITS and plastid markers (Bozek et al. 2012: https ://doi.org/10.1111/j.1095-8339.2012.01314.x) failed to show any resolution among species of L. sect. Luzula, which is not surprising, given their close relationships. In the frame of the bigger project LuzAlp (https://www.uibk.ac.at/de/botany/forschung/projekte/same-same-but-different/) that is currently running we will perform RADseq and whole plastome sequencing and thus hope to obtain further insights in the diversification of L. sect. Luzula.
Round 2
Reviewer 1 Report
Dear authors,
Thank you for your notes. I consider it is necessary to reject the article until the authors submit digitized images of voucher specimens.
Author Response
Unfortunately, as stated before, we are not able to provide digitised images at this moment. In addition, digitised images are neither required by this journal nor by and other as a requirement for a paper to be published. In our opinion, this should be handled in the same way for all papers published.